# Colouring the Higgs boson

**Andy Buckley[1], Giuseppe Callea[1*], Andrew J. Larkowski[2] and Simone Marzani[3]**

**1** School of Physics and Astronomy, University of Glasgow, Glasgow G12 8QQ, Scotland, UK
**2** Physics Department, Reed College, Portland, OR 97202, USA
**3** Dipartimento di Fisica, Università di Genova and INFN, Sezione di Genova,
Via Dodecaneso 33, 16146, Italy

⋆ giuseppe.callea@cern.ch

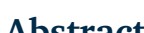
## Abstract

**The jet colour ring is a novel colour tagger observable designed to separate the decay of a colour-singlet into two jets from a two-jet background in a different colour configuration. Simulation studies in the case of the production of a boosted Higgs boson decaying in two b-quarks and an associate electroweak boson, showed notable discriminator powers when comparing the jet colour ring performances with other observables. These results are opening a wide scenario for further studies.**

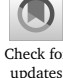

## 1  Introduction

Studying the internal structure of hadronic jets is crucial in the context of new physics searches and Standard Model constraints. Observables sensitive to the colour configuration of near-collinear pairs of jets are expected to show a notable discrimination power. In this contribution, a new colour-sensitive observable called the jet colour ring [1] is introduced. Its performances were assessed by studying the decay of a colour-singlet, like the decay of the Higgs bosons into (bottom-quark) jets. The results were compared to other colour-sensitive observables, called the pull angle [2] and dipolarity [3]. Given these assumptions, the jet colour ring outperforms these observables in discriminating the colour-singlet configurations.

## 2  The colour ring

The idea behind the design of the colour ring is rather simple. By the Neyman-Pearson lemma, the optimal discriminant observable for distinguishing the momentum dependence of these

colour configurations is monotonic in their likelihood ratio. Therefore, the ratio of the soft gluon emission background ($|M_B|^2$) and the colour-singlet signal ($|M_S|^2$) matrix elements is considered by exploiting the kinematics of the dipole configuration in the collinear limit. Eq.1 shows the new discriminating variable in the collinear limit, where $\theta_{ak}$ ($\theta_{kb}$) is the angle between the soft gluon and each of the final-state hard partons, and $\theta_{ab}$ is the angle between them.

$$\frac{|M_B|^2}{|M_S|^2} \simeq \frac{1-\cos\theta_{ak}+1-\cos\theta_{bk}}{1-\cos\theta_{ab}} \to \mathcal{O} = \frac{\theta_{ak}^2+\theta_{bk}^2}{\theta_{ab}^2}. \tag{1}$$

The jet colour ring expression, $\mathcal{O}$, is given by Taylor expanding the cosine in Eq.1. $\mathcal{O}$ can take values greater or less than 1, depending on the location of soft emission k. To better interpret this expression, a geometric illustration of $\mathcal{O}$ is shown in Fig.1. $\mathcal{O}$ is equal to 1 for the circle with diameter equal to $\theta_{ab}$, where the two antipodal points correspond to the bottom and anti-bottom quark directions. Emissions from a colour-singlet dipole are dominant within the $\mathcal{O}=1$ circle, while emissions from a colour-octet dipole dominantly lie outside it.

For the studies considered in this contribution, the signal and background processes are H($b\bar{b}$) + g and g$\to b\bar{b}$ + g, respectively, where the extra gluon is the soft real-emission parton k.

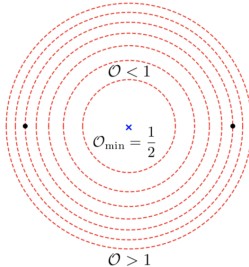

Figure 1: Illustration of the geometry selected for by the observable $\mathcal{O}$. The location of the final-state hard partons are denoted by the black dots, and the $\mathcal{O}=1$ contour passes through them. Figure reproduced from [1], used under CC-BY license.

## 3 Results

The performances of the jet colour ring were investigated in Monte Carlo simulations and compared against the aforementioned pull angle and dipolarity, with the addition of the $D_2$ observable [4]. MADGRAPH5_AMC@NLO 2.6.7 in LO mode, with parton showers and non-perturbative effects provided by Pythia 8 [5] was employed for the simulation of both signal and background process. The Rivet 3 toolkit [6] was used in the event analysis. Flavour-tagged small-R track-jets, matched to a high-$p_T$ large-R jet, are used to identify the two b-tagged jets and a third light-jet.

Using track-jets presents particular advantages, as they provide a high-resolution estimate of the orientation and size of the radiating dipole within a jet. Fig.2a shows the colour ring distribution for the Z($\mu\mu$)H($b\bar{b}$) and Z($\mu\mu$)+$b\bar{b}$ contributions. As expected, the colour-singlet contribution mostly populates the $\mathcal{O}$ < 1 region and peaks at low $\mathcal{O}$ values. Fig.2b shows the ROC curves of the four observables considered. These curves characterise the trade-off between the true-positive event identification rate, and the false-positive identification rate. While the colour ring outperforms both pull angle and dipolarity, it is suboptimal compared to $D_2$. As both $D_2$ and $\mathcal{O}$ use different strategies to probe the colour configuration of the decaying particle, it is natural to combine them as they contain significant orthogonal information. Moreover, the colour ring adds awareness of the orientation and opening of the $b\bar{b}$

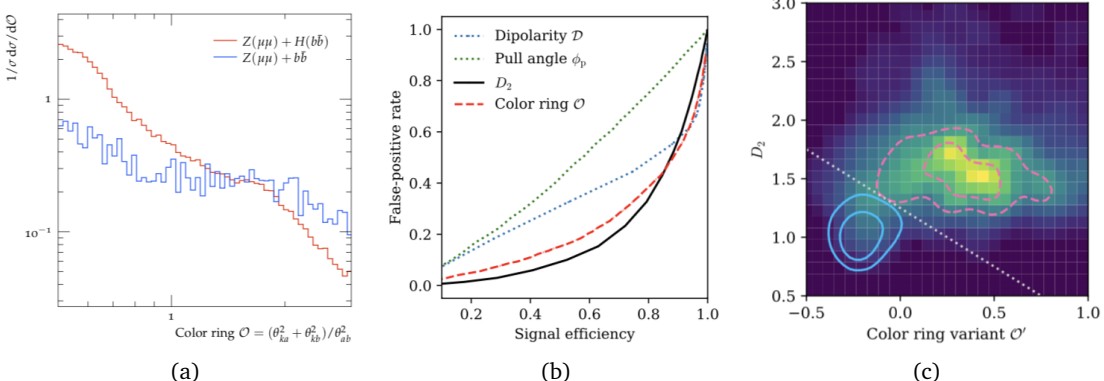

(a)    (b)    (c)

Figure 2: Fig.2a shows the $Z(\mu\mu)H(b\bar{b})$ vs. $Z(\mu\mu)b\bar{b}$ distributions for the colour ring observable. Fig.2b shows the ROC curves for the four observables considered in this study. Fig.2c illustrates the complementarity of $D_2$ with the colour ring in a 2D distribution,with contour overlays at 50% and 75% of the maximum values for signal (solid) and background (dashed) densities separately. Figure reproduced from [1], used under CC-BY license.

dipole. Hence, greater background rejection for any signal efficiency is possible by using a two-dimensional cut in the plane. The colour ring and the combined variable could have a significant impact if introduced in the $H(b\bar{b})$ MVA analyses.

## 4 Conclusion

The jet colour ring is a new observable sensitive to colour-singlet configurations. Instead of studying the radiation pattern of the signal and backgrounds processes to separate them, it simply attempts to build a tagger from the ratio of their matrix elements in the collinear limit. Notable discrimination power was observed in the context of the Higgs boson decay into bottom quarks. Moreover, it offers complementary information to the energy correlation functions based on jet shapes, such as $D_2$. Further theoretical and experimental refinements are possible giving this simple idea and it would be interesting to integrate the jet colour ring in experimental studies. The boosted VH(bb) analysis [7] would be an interesting testing ground for its discrimination power. An alternative, track-jet $p_T$ weighted, version of the colour ring is under consideration. This approach improves the applicability of the jet colour ring and allows some cancellation on the overall track-jet $p_T$ uncertainties. Preliminary results of the new colour ring version are shown in Fig.3

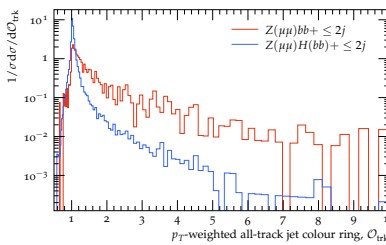

Figure 3: Alternative track-jet $p_T$ weighted colour ring. This version does not require the presence of three track-jets in the event and improves its applicability.

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
