# Peer review of "Colouring the Higgs boson"

_SciPost Physics Proceedings, doi:SciPost Phys. Proc. 10, 001 (2022)_

## Round 2 · Referee Report · Anonymous (Referee 1) · 2022-1-7

Report

In this contribution, the authors introduce the jet color ring, as a novel color-sensitive observable, useful for the discrimination of boosted massive color-singlet into two jets, from the background.
The observable is derived from the ratio of the squared amplitude of the background process, namely the soft gluon emission, w.r.t. one of the signal, namely the color-singlet decay. Within the small angle approximation, the jet color ring happens to have a remarkably simple formula. The nature of the decaying particle is reconstructed from the location of the final-state hard partons w.r.t the regions of the ring: emissions from a color-singlet dipole would populate the inner region, while the ones from a color-octet, the outer regions.
The proposed observable can improve the strategies for jet tagging, in order to providing the estimation of the geometry of the radiating dipole within a jet at higher-resolution.
The authors successfully apply their novel strategy to the decay of the Higgs boson into bottom anti-bottom jets.

The presentation is very interesting and deserves publication on SciPost.

As a minor comment, let me suggest the authors add a comment about the infrared safety of their observable, which interested readers may find it of relevance.

---

## Editorial Decision

published